# Education Makes the Difference: Work Preferences of Young Adults with Physical Disability

**DOI:** 10.3390/ijerph19159021

**Published:** 2022-07-25

**Authors:** Limor Gadot, Yifat Faran, Orly Sarid

**Affiliations:** 1School of Social Work, Sapir Academic College, Sha’ar HaNegev Regional Council 7956000, Israel; 2Faculty of Social Work, Ashkelon Academic College, Ashkelon 78211, Israel; yfi@post.bgu.ac.il; 3Spitzer Department of Social Work, Faculty of Humanities and Social Sciences, Ben-Gurion University of the Negev, Beer Sheva 84105, Israel; orlysa@bgu.ac.il

**Keywords:** physical disabilities, young adult, work, education, social factors, social norms

## Abstract

This study focused on the work preferences of young adults with physical disabilities (YAPD) in Israel and the variables that affect those preferences. The theory of planned behavior (TPB) was employed to explain work preferences. We examined direct and indirect links between education and socioeconomic status (SES) in a comprehensive model that tested the mediating role of the TPB and self-assessed health. A cross-sectional study was conducted throughout 2017. Participants included 348 YAPD aged 18–30 not yet integrated into the workforce. Exploratory factor analysis of work preferences yielded three dependent variables: ‘intention to work’, ‘interest, security, and advancement at work’, and ‘willingness to work in the free market’. Data analysis included correlations and path analysis by structural equation modeling. Education was positively associated with all work preferences, while SES was positively associated with ‘interest, security, and advancement at work’. Subjective norms mediated the relation between education and ‘intention to work’ and ‘interest, security, and advancement at work’. Self-efficacy mediated the relation between SES and ‘interest, security, and advancement at work’. Finally, self-assessed health mediated between SES and self-efficacy. Education is a crucial human capital in predicting work preferences of YAPD. The TPB components are important factors in predicting work preferences.

## 1. Introduction

The current study focuses on young adults with disability (YAPD), their intention to work, and their work preferences. Specifically, we examined the direct and indirect links between sociodemographic variables, health-related variables, and intention to work.

## 2. Literature Review

Over the last decades, society’s perceptions of the rights of people with disabilities have changed. One change has to do with the transition from a medicalization approach to a social functional approach which stresses the importance of social inclusion [1]. Several benefits of participating in the workforce, in addition to inclusion, have been acknowledged in the literature, including enhancement of socioeconomic status (SES) [2], and acquiring social and practical skills such as teamwork and customer service, problem-solving, and communication [3]. Several studies have also shown that being employed fosters physical and psychological wellbeing in the general population [4], and particularly among people with disabilities [5].

A major factor that influenced the change in laws and rights regarding the inclusion of people with disabilities in the labor force was the UN Convention on the Rights of Persons with Disabilities (CRPD) [6]. According to Article 27(1) of the CRPD, the right of people with disabilities to work is equal to all other people and includes the right to make a living by working in a work environment and labor market that are open, inclusive, and accessible to persons with disabilities [7].

Following the CRPD, many Western countries including Israel have tried to implement policies that incentivize work and promote the workforce integration of people with disabilities, in parallel with policies aimed to reduce welfare dependency [8,9,10]. Some of these policies target young adults with physical disabilities (YAPD). Indeed, previous studies among YAPD have shown that early integration in the workforce helps develop long-term career plans and enhance a sense of productivity and social integration [11,12]. Therefore, this study uses a representative sample of YAPD across Israel.

In Israel, there are 1,493,100 people with disabilities: approximately 15% (704,300) are working age adults (15–64), and 230,000 receive a disability allowance. Current data indicate that only 51% of them are employed, compared to 79% of their non-disabled counterparts [13]. These rates are similar to those in other Western countries. For instance, in the US, the employment-to-population ratio is 30.5% and 73.5% for people with and without disabilities, respectively [14]. Similarly, in Canada, the respective ratios are 48% and 76% [9]. Finally, in the EU, only a minority of people who receive a disability allowance are employed in the open market, despite integrative government policies [8].

A major policy practice for increasing the labor market integration of people with disabilities is providing vocational rehabilitation, including higher education. In Israel, the National Insurance Institute (NII) is responsible for benefits such as disability allowance and vocational rehabilitation for people with disabilities [15]. According to the NII, a person with a disability can receive vocational rehabilitation, including higher education, if they meet certain criteria. These include diagnosis by a medical committee of the NII and being a resident of Israel who has turned 18 and has not yet reached retirement age [16], who, due to functional consequences arising from the disability, is unable to return to work or requires vocational training to integrate into the labor market [16]. Thus, people with disabilities are defined as “able” or “unable” to work in the free market, with the former being entitled to vocational rehabilitation.

Given the importance of the labor market on wellbeing for people with disabilities, it is crucial to assess YAPD, their work preference, and intentions to participate in the workforce, as well as personal factors and social factors that affect these intentions and preferences. Previous studies have found high and positive associations between preferences and intentions with respect to career decisions [17,18].

### 2.1. Theory of Planned Behavior (TPB)

One of the main theories that focus on motivational factors predicting intentions is the theory of planned behavior (TPB). The TPB focuses on internalized mental aspects, such as attitudes, subjective norms, and perceived control, and their effect on the intention to carry out behavior [19]—in our case, intention to work and work preferences of YAPD. Attitudes towards a behavior are perceptions and beliefs that a particular behavior will lead to a particular result. They may be positive or negative. Subjective norms reveal the beliefs of individuals about how they would be viewed by their peer group or family if they perform a certain behavior [20]. The TPB was applied to examine health intentions [21]. More specifically, it was applied to examine intentions associated with work, for example, among young adults with physical disability and their behavioral intention to travel to work independently [22]. It was also used to investigate behavioral intentions such as job search [23], and work preferences among health professionals [24]. Among people with severe mental disorders enrolled in supported employment programs, the TPB explains intentions for and actual integration in competitive jobs [25].

Finally, perceived behavioral control refers to people’s perceptions of their ability to perform a given behavior. It is a distinct concept of motivation, and according to Ajzen [26], it is related to self-efficacy. Self-efficacy is defined as beliefs we have in our own abilities, specifically our ability to meet the challenges ahead of us and complete a task successfully [27]. Indeed, self-efficacy scores have been shown to be the best predictors of career interests among people with a learning disability [28]. A study among people with physical disabilities found a positive correlation between self-efficacy and return to work after a long absence [29]. In addition, in previous studies among people with chronic illness, positive correlations were found between self-efficacy and self-assessed health [30,31].

Other variables that are related to self-efficacy and are not addressed by the TPB are work experience, education, and SES. The relation between work experience and intention to work has been studied, with work experience among people with disability linked to successful work integration [32] and job placement [33]. For example, a study of young adults with autism in Israel found that work experiences were related to high pay [34]. Education was also tested [14,35], and low education was associated with difficulties to find well-paying jobs [14] and with preferences to work in SMEs [36]. To the best of our knowledge, SES was less studied and there no studies testing the link between education and SES, and intention to work and work preferences. Therefore, our first aim was to examine the direct links between work experience, education, and SES on the one hand, and intention to work and work preferences of YAPD on the other.

### 2.2. Mediating Role of Self-Assessed Health

Among people with disabilities, self-assessed health was found to play an active role in behaviors, such as participation in the workforce [37]. High self-assessed health was associated with willingness to return to the workforce, whereas low self-assessed health was associated with unemployment [38]. Among people with disabilities, self-assessed health mitigated the links between education and willingness to be employed part time. Furthermore, the ability to obtain and retain a high-quality job was related to perceived health status, rather than to the degree of disability [39]. As maintained earlier, self-assessed health was related to self-efficacy [30,31]. Therefore, our second aim was to examine the mediation effect of self-assessed health on the link between work experience, education and SES, and self-efficacy. Finally, we examined, in a comprehensive model, the mediating effect of the TPB components (attitudes toward work, subjective norms, and self-efficacy) and self-assessed health on the link between work experience, education, and SES, and intention to work and work preferences.

We hypothesized that work experience, education, and SES would be positively and directly associated with intentions to work and work preferences. We further hypothesized that self-assessed health would mediate the relation between work experience, SES, education, and self-efficacy to intentions to work and work preferences. Finally, we hypothesized that the TPB components would mediate the link between work experience, SES, and education, and intentions to work and work preferences.

## 3. Materials and Methods

### 3.1. Participants and Procedures

A cross-sectional study was conducted among 348 YAPD in Israel (185 women). Data were gathered from November 2016 to November 2017. The inclusion criteria were adults with a physical disability recognized by the National Insurance Institute, aged 18–30, Hebrew-speaking, and with at least 12 years of schooling. Excluded were people with mental disability or intellectual disability. The sample size was calculated using the G *Power 3.1 software, Heinrich Heine University Düsseldorf, Düsseldorf, Germany, to ensure sufficient statistical power [40]. The research model included 14 adapters. In a pretest of 37 people with disabilities, 9 adapters were found to be strong. To support the hypotheses in at least five correlations and based on the following criteria for the researcher’s hypothesis: α = 0.01 and a power of 1 − β = 0.95 (1 − β = 0.99 per coordinate), a sample of N = 342 was required.

We assumed an 8% dropout rate, and thus approached 382 individuals, and the 348 who completed the questionnaire constituted the final sample. The response rate was 92.6%, similar to that reported by Ivzori [37], who examined the path to working life among young Israelis with developmental intellectual disabilities. Previous studies among YAPD reported a lower response rate of 72% [41].

Following ethical approval by the Ministry of Labor, Social Affairs, and Social Services and the Ministry of Education, letters were sent to ten rehabilitation centers and four special education schools relevant to the research population, located across Israel. Two schools and six rehabilitation centers agreed to participate and administer the questionnaires. We also sent letters to all public colleges and universities in Israel. Out of 28 public academic institutes, about a third agreed to administer the questionnaire. Participation in the study was voluntary and anonymous. Participants could stop completing the questionnaire at any time and they were assured of anonymity.

The participants’ ages ranged from 18 to 30 (MAGE = 24.5, SD = 3.51, range 18–30).

Table 1 presents the participants’ demographic variables.

As can be seen from Table 1, 167 (48%) of our participants had high school education, and 124 (36%) reported good SES. Most of them were single—293 (84%)—and lived with their origin family. Almost half did not work at all (161, 46%). Most often, the reported disability was related to the nervous system (48, 14%), and disability types were classified according to NII criteria [42]. Most of our participants had a congenital disability (229, 65%).

### 3.2. Measures

#### 3.2.1. Dependent Variables

Intention to work and work preferences were measured using three questionnaires. The first, based on the TPB Questionnaire [19], included four items; for example, “How much are you interested in working when you graduate from school/national or military service/university?” The second included seven items [43]; for example, “How much would you prefer your job to be interesting?” The third questionnaire [14] comprised six items, such as “How much would you want to work full time?” Three questions were added to assess intentions regarding work environment, such as “How much would you prefer to work in a supported employment?” All responses were recorded on a 5-point Likert scale from 1 (“not at all”) to 5 (“very much”).

Based on answers from the work preferences questionnaires, we conducted an exploratory factor analysis (EFA) with varimax rotation [44] using the initial eigenvalue cutoffs. The factor solution yielded three factors with eigenvalues above 1.0, which included at least three items. Table 2 presents the factor loadings.

As can be seen from Table 2, the exploratory factor analysis yielded three factors that included three items or more. Factors that included less than three indications from the questionnaire were excluded from further analysis. The three factors were ‘intention to work’, ‘interest, security, and advance at work’, and ‘willingness to work in the free market. The cumulative variance explained by these factors was 34.32%. For each factor, a mean score of the relevant questions was calculated for each participant, with higher scores indicating higher ‘intention to work’, ‘greater desire to achieve interest, security, and advancement at work’, and higher ‘willingness to work in the free market’.

#### 3.2.2. Independent Variables

The independent variables included demographic data: age, gender, education (high school, vocational training, higher education studies, BA degree or higher), SES (extremely low, low, fair, very high, extremely high), and work experience (not employed, volunteer, sheltered employment, supported employment, and working in the labor market (paid job)).

#### 3.2.3. Mediating Variables

Attitudes towards work were measured using ten items from Kanungo’s [45] Job Involvement Measure; for example, “I consider my job to be very central to my existence” and “I like to be absorbed in my job most of the time”. Responses were rated on a 5-point Likert scale from 1 (“totally disagree”) to 5 (“very much agree”). The questionnaire was translated into Hebrew and was found to have high construct validity among ultraorthodox women [43,46]. Mean scores were calculated, with a high score indicating higher perceived centrality of work. Cronbach’s alpha = 0.73.

Subjective norms were measured using Dunstan et al.’s [47] three-item scale for family members, friends, and peers; for example, “My family members who are important to me think I should decide to go to work”. Responses were rated on a 5-point Likert scale from 1 (“totally disagree”) to 5 (“very much agree”). Mean scores were calculated, with a high score indicating a higher level of subjective norms. Cronbach’s alpha = 0.83.

The self-efficacy questionnaire [27] included eight statements, such as “I will be able to achieve most of the goals that I have set for myself”. Responses were rated on a 5-point Likert scale, from 1 (“totally disagree”) to 5 (“very much agree”). Mean scores were calculated, with a high score indicating higher self-efficacy. Cronbach’s alpha = 0.914. The questionnaire was used in Israel among young adults with autism [48]. Perceived behavior control [20] included three questions; for example, “For me, choosing a workplace depends only on me”. Responses were rated on a 5-point Likert scale from 1 (“totally disagree”) to 5 (“very much agree”). Mean scores were calculated, with a high score indicating higher perceived behavior control. We combined the two questionnaires because together they provided a high degree of internal reliability in the present study: Cronbach alpha = 0.83.

Self-assessed health was measured using the Short-Form Survey (SF-12) [49]—a 12-item questionnaire that provides a shorter alternative to the SF-36. It includes two summary score components: physical (6 items), for example, “During the past 4 weeks, have you had any of the following problems with your work or other regular daily activities as a result of your physical problems?”, and mental (6 items), for example, “During the past 4 weeks, have you had any of the following problems with your work or other regular daily activities as a result of any emotional problems (such as feeling depressed or anxious)?” Each item is scored on a scale of 0 to 100. Higher scores indicate better health. Cronbach’s alpha = 0.82. The questionnaire was translated into Hebrew and was found to be valid and reliable among young adults with disability: Cronbach’s alpha = 0.707 [37].

### 3.3. Data Analysis

The data were analyzed using SPSS version 27.0 and an AMOS module for structural equation analysis (Armonk, NY, USA: IBM Corp). Means and SDs for each variable were calculated, and Pearson’s correlations between all variables were calculated as well. The potential for multicollinearity was addressed by calculating the variance inflation factor (VIF) values for each independent variable. All VIF values were close to 1, so that multicollinearity was ruled out [50]. EFA was conducted on all the work preferences items, aiming to find distinct aspects of work preferences, as described in the method section (Table 2).

The IBM SPSS AMOS (27) program for assessing structural models [51] was used to examine both direct and indirect effects among the TPB variables, including those not included in the original model. The predictors were education and SES, and age, gender, and work experience showed no significant relations to any of the mediating or dependent variables and were thus excluded. The mediators had two layers: self-assessed health in the first and attitudes toward work, self-efficacy, and subjective norms in the second. The predictors were ‘intention to work’, ‘interest, security, and advancement at work’, and ‘willingness to work in the free market’. Mediation relationships observed in the path analysis were subsequently tested using indirect analysis [51] to examine which of the mediated relationships were significant.

## 4. Results

Means, SDs, intercorrelations, and VIFs are presented in Table 3.

As can be seen from Table 3, age was associated only with ‘intention to work’. Gender was not associated with any of the predicted variables. Work experience was associated with ‘intention to work’ and ‘willingness to work in the free market’. Education was associated with ‘intention to work’, ‘interest, security, and advancement at work’, and ‘willingness to work in the free market’. SES was associated only with ‘interest, security, and advancement at work’. Self-assessed health was not associated with any of the work preferences. Among the TPB-mediating variables, self-efficacy and subjective norms were associated with ‘intention to work’, ‘interest, security, and advancement at work’, and ‘willingness to work in the free market’. Attitudes toward work were associated with intention to work and with interest, security, and advancement at work. None of the correlations between the predictors or the mediators were below 0.7, indicating no multicollinearity, and VIF values supported this finding.

### Path Analysis by Structural Equation Modeling

Structural equation modeling (SEM) included four layers of variables: predictor variables, mediation of mediating variables, mediating variables, and outcome variables. It revealed a good fit between the model and data (*N* = 354; χ^2^ = 12.423, df = 11, *p* = 0.333). The NFI and CFI values were 0.980 and 0.998, respectively, and the RMSEA value was 0.019, also indicating good fit [51]. Note that work experience, age, and gender did not contribute significantly to any of the variables and were therefore excluded from the model.

As can be seen from Figure 1, self-efficacy, subjective norms, and attitudes toward work mediated the relations from education and SES to all three dependent variables. Moreover, self-assessed health mediated the relations from education and SES to some of the mediators and some of the dependent variables. Mediations revealed by the path analysis model were examined more closely by means of indirect analysis [52]. Only mediation relationships that were statistically significant according to the path analysis were reexamined in the indirect analyses.

As can be seen from Table 4, confidence interval (CI) values indicated significant results for each of the mediated relationships, as their lower and upper bounds were not crossed by the value of zero [52]. *Z* of Sobel [53] was also used to evaluate the significance of the same mediated relationships. As can be seen, some of the mediating relations predicted were found to be significant.

The SEM produced three direct positive links from education to the three outcome variables: ‘intention to work’, ‘interest, security, and advancement at work’, and ‘willingness to work in the free market’, whereas SES was not directly related to any of the outcome variables. This finding supports our first hypothesis regarding education, but not SES. We also found 15 mediation relations. Self-assessed health mediated the relation between education and SES to self-efficacy, supporting our second hypothesis. Self-assessed health was also positively associated with ‘interest, security, and advancement at work’.

Self-efficacy mediated the relation to the three outcome variables. Education was negatively associated with self-assessed health, which in turn was positively associated with self-efficacy, which was positively associated with the three outcome variables. For these mediations, indirect analysis [51] revealed that only the effects of education on ‘intention to work’ and ‘interest, security, and advancement at work’ were mediated via subjective norms. In both cases, the mediation effect increased the total effect of education on the dependent variable. In other words, part of the contribution of education to ‘interest, security, and advancement at work’ was due to its influence on subjective norms. Similarly, part of the contribution of education to ‘intention to work’ was due to its influence on subjective norms.

SES was positively associated with self-efficacy. Thus, self-efficacy mediated the relation between SES and the three outcome variables. Subjective norms mediated the relations between education and ‘intention to work’ and ‘interest, security, and advancement at work’. Subjective norms also had a mediation effect on the relation between SES and ‘intention to work’ and ‘interest, security, and advancement at work’. Finally, the relation between SES and ‘interest, security, and advancement at work’ and ‘intention to work’ was mediated by attitudes toward work. Indirect analysis [52] of these mediations revealed that only the effect of SES on ‘interest, security, and advancement at work’ via self-efficacy was significant. This mediating effect increased the total effect of SES on ‘interest, security, and advancement at work’. In other words, part of the contribution of SES to ‘interest, security, and advancement at work’ was due to its influence on self-efficacy.

Table 5 presents the mediation effect of self-assessed health on the mediators and predictors.

As can be seen from Table 5, the indirect analysis revealed a significant mediation pathway between SES and self-efficacy, with self-assessed health as the mediator. In other words, part of the contribution of SES to self-efficacy was due to its influence on self-assessed health. Thus, our last hypothesis was supported, apart from the fact that attitudes towards work did not mediate the relation between education and outcome variables.

## 5. Discussion

The aim of the current study was to examine direct and indirect links from work experience, education, and SES to intention to work and work preferences of YAPD. In addition, we examined the mediating role of self-assessed health on the relations between work experience, education, and SES and the mediators and outcome variables. Finally, we tested the mediating effect of attitudes, subjective norms, and self-efficacy, as proposed by the TPB model, on the links between the work experience, education, and SES, and intention to work and work preferences.

We hypothesized that education, SES, and work experience would be positively and directly associated with behavioral intentions to work and work preferences of YAPD. Our findings partly confirmed our hypothesis. Higher levels of education were associated with higher ‘intention to work’, ‘interest, security, and advancement at work’, and ‘willingness to work in the free market’, as presented in the SEM model. These findings are in contrast with another study that found that academic qualifications did not affect the probability of employment [15] but are consistent with another study that found that education predicted intentions and preferences to study abroad as part of academic progress [54]. Our findings expand on these by showing that education positively and directly influences the intention to work and work preferences of YAPD.

Contrary to our prediction, as observed in the SEM model, SES and work experience were not related to intention to work and work preferences. However, the SEM model showed that self-assessed health and self-efficacy mediated the link between SES and the preference of ‘interest, security, and advancement at work’. Our findings corroborate those of Conner et al. [55], who observed no direct pathways between SES and the intention to avoid smoking. However, findings of another study showed a direct link between SES and the intention to drop out of high school [56]. A possible explanation for the incongruity of the above findings is that most of the YAPD participating in our study lived at home and were dependent on their parental SES, and therefore they did not consider SES as a crucial factor affecting their work preferences. Furthermore, most of the participants had higher SES, and therefore lived in a relatively protected environment. This can explain, at least partly, the lack of a direct effect.

Following the TPB model, we examined whether attitudes, subjective norms, and self-efficacy mediated the links from the predictor variables to intention to work and work preferences. Self-efficacy was related to ‘willingness to work in the free market’ but testing the mediation contribution of this variable using PROCESS did not indicate mediation of the pathway between SES and this work preference. A possible explanation for this finding is that self-efficacy is an internal factor that leads to behavioral intentions, but among our young participants who lived with their origin families and depend on their parents, this internal resource was not fully developed, and it did not serve as a motivational component that could link SES and work preferences. We also did not find any contribution of work experience to the outcome variables. This contradicts previous studies [32,34]. Recall, however, that our participants were young and most of them did not have rich working experience.

Our second hypothesis was that self-assessed health would mediate the relation between work experience, SES, education, and self-efficacy, and intentions to work and work preferences. This hypothesis was fully confirmed. The TPB model does not include the component of self-assessed health. However, among people with disabilities, this factor is an essential mediating factor. These findings point at the significance of ones’ health perception in predicting YAPD’s working ambitions and capability [37,39].

Finally, we hypothesized that the TPB components (attitudes toward work, subjective norms, and self-efficacy) would mediate the link between work experience, SES, and education, and intentions to work and work preferences. This hypothesis was partly confirmed. Work experience did not contribute to predicting the outcome variables, neither directly nor through mediation. In addition, we found that subjective norms and self-efficacy mediated the relations from education and SES to ‘intention to work’ and ‘interest, security, and advancement at work’. However, attitudes towards work did not mediate the relations from education, SES, and work experience to intention to work and work preferences. Previous studies among people with disabilities found that attitude toward work served as a mediator between background characteristics and the intention to return to work [25,29]. A possible explanation for our finding is that our participants were young and did not actually work. In that case, their attitudes might be based on the attitudes acquired in their families and on their minor experiences at work.

In the current study, we used the TPB model to predict YAPD’s intention to work and work preferences. Implementing this model to intention to work and work preferences of YAPD is novel. In the specific case under study, it showed that similar mechanisms predict health intentions and work preferences. The mediating components defined by Ajzen [20] are also important in predicting health intentions and behaviors as well as in predicting intention to work and work preferences, specifically, subjective norms and self-efficacy. However, not all TPB components contributed to the same degree to predicting the outcome variables. That is, subjective norms mediated the relation from education and SES to work preferences, and self-efficacy mediated the relation from SES to interest, security, and advancement at work. However, attitudes towards work did not mediate any of the relations. Thus, it seems that subjective norms are an essential component and influence work preferences of YAPD. Finally, a contribution of our study to the TPB model is the monitoring of self-assessed health as a mediator between the education, SES, and self-efficacy of YAPD.

Our research suggests several policy implications. We have found that higher education is a crucial human resource related to intention to work and work preferences. However, higher education for people with disability in Israel depends largely on the NII: people with disabilities are defined as “able” or “unable” to work in the free market. The latter are not entitled to higher education as part of their eligibility for vocational rehabilitation. We suggest that the option of subsidized higher education should be open to all YAPD, and thus serve as an incentive to increase the intention to work and willingness to enter the workforce. Furthermore, higher education would open up options for work preferences for YAWD.

A second policy implication is related to the TPB component and its translation to policymakers. We found that subjective norms are a crucial factor in predicting intention to work and work preference. Since the family is a major socialization agent, subjective norms are also shaped within the family [57]. We recommend that rehabilitation services develop and implement programs for families of YAPD as part of employment programs during and after high school. Socialization of family members to the possible integration of their children in the workforce can contribute to changing work-related motivations.

Finally, we believe that intention to work and work preferences should be cultivated among all children, and especially for children with disabilities, from a very early age. This socialization stresses the productive role of the individual in society, regardless of the disability barriers.

### Limitations and Future Directions

Together with its merits, this study has several limitations. The first is the recruitment method: participants entered if they met the inclusion criterion of age 18–30. Since attitudes and self-efficacy are personal resources that develop throughout life, we strongly encourage future studies to recruit younger YAPD and monitor their personal resources across several years. We also recommend that future studies recruit older people with disability and focus on the effect of SES as personal resources on intention to work and work preferences The second limitation is that other disabilities, such as mental health disabilities, were not included in this study. Future research can replicate our study with young adults with mental health disabilities. The third limitation is that we focused on the Jewish population. Future studies should include YAPD from other populations in Israel. Fourth, this study included only people with high school education and above, yet a substantial population of YAPD exists in Israel which has not graduated from high school. Accordingly, we recommend that future research include this population as well. We also recommended adding a question about the participants’ vocational certificate. Finally, the cross-sectional design did not enable us to follow our participants over the long term to examine whether their preferences were fulfilled. Future research is required to follow YAPD across time, also considering attitudes and beliefs of their parents and siblings.

## 6. Conclusions

This study focused on the work preferences of young adults with physical disabilities (YAPD) in Israel and the variables that affect those preferences. The theory of planned behavior (TPB) was employed to explain work preferences. This study presented the importance of education and SES to work preference of YAPD. It demonstrated the mediating role of self-efficacy, subjective norms, and attitude towards work (TPB components) in predicting work preferences. As subjective norms play an important role in influencing intention to work and work preferences, future research should explore this determinant. In addition, the model included self-assessed health and showed that among YAPD, it mediated the relation between education and socioeconomic status as well as the TPB components and outcome variables. Our findings indicate that even elements that are ostensibly related to personal factors, such as self-efficacy and self-assessed health, are affected by social factors. Our findings correspond with previous studies that addressed the effects of the environment and society on internal motivational factors of people with disabilities. By no means do we relieve society from its responsibility to open and promote work options to people with disabilities. Consequently, policymakers need to take these findings into consideration, especially the realization that young adults with disabilities should receive higher education, and this in turn is linked with their intention to participate in the workforce.

## Figures and Tables

**Figure 1 ijerph-19-09021-f001:**
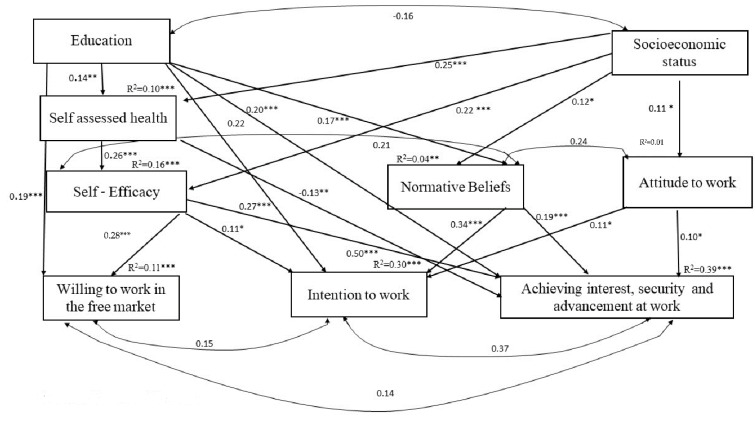
The path analysis model with significant effects and correlations (i.e., standardized path coefficients) between the variables. * *p* < 0.05, ** *p* < 0.01, *** *p* < 0.001.

**Table 1 ijerph-19-09021-t001:** Participants’ demographic variables.

Variable	N (%)
Age	24.5 (SD = 3.51)
Gender (women)	185 (53)
Education	
High school	167 (48)
Professional Training	28 (8)
Academic studies	107 (31)
Academic degree (BA)	45 (13)
SES	
Extremely not good	14 (4)
Not good	25 (7)
Fair	120 (34)
Good	124 (36)
Extremely good	65 (19)
Marital status	
Married/in a relationship	56 (16)
Single	293 (84)
Residence	
Independent residence	131 (38)
Living with family of origin	293 (84)
Work experience	
Not working	161 (46)
National service	43 (12)
Sheltered employment	46 (13)
Supported employment	23 (7)
Labor market	75 (22)
Type of disability	
Nervous system	160 (48.2)
Skeletal and muscular	49 (14.8)
Hearing	43 (13)
Vision	28 (8.4)
Chronic illness	25 (7.5)
Genetic disorder	23 (6.9)
Other	4 (1.2)
Onset of disability	
Congenital disability	229 (65)

Note: Participants’ ages ranged from 18 to 30.

**Table 2 ijerph-19-09021-t002:** Factor loading by EFA (pattern matrix of factors and items).

	Interest, Security, and Advancement at Work	Intention to Work	Willingness to Work in the Free Market	Part Time vs. Full Time	Organization Size
How much would you prefer an interesting job?	**0.815**	0.058	0.168	−0.090	−0.002
How much would you prefer a job where you can exercise your talents?	**0.761**	0.302	0.077	0.001	−0.048
How much would you prefer opportunities for advancement?	**0.760**	0.116	0.031	0.072	0.107
How much would you prefer a job with fair wages?	**0.731**	0.337	0.059	0.001	−0.052
How much would you prefer a job with responsibilities?	**0.627**	0.116	0.073	0.236	0.104
How much would you prefer working independently?	0.590	0.145	0.007	0.110	−0.241
How much would you prefer a job with tenure/occupational security?	**0.584**	0.330	0.048	0.004	0.075
How much would you prefer a job with social security benefits?	0.479	0.440	0.024	−0.271	0.104
How much are you interested in going to work in the next 5 years?	0.241	**0.702**	0.131	0.007	−0.021
How much do you estimate that your friends, who are your age, will go out to work in the future?	0.121	**0.698**	0.066	0.187	0.010
How much are you interested in working when you graduate from school/national or military service/university?	0.311	**0.686**	0.126	0.007	0.112
How much do you have good friends who are of working age and working?	0.165	**0.650**	−0.027	0.043	−0.125
How much would you prefer to work in sheltered employment?	−0.078	−0.142	**−0.749**	0.012	0.075
How much would you prefer to work in the free market without support?	0.275	0.172	**0.707**	0.027	−0.030
How much would you prefer to work in supported employment?	0.055	0.083	**−0.659**	−0.096	−0.184
How much would you prefer to work full time?	0.056	0.016	−0.038	**−0.874**	−0.040
How much would you prefer to work part time?	0.205	0.215	0.077	**0.811**	0.051
How much would you prefer to work in a small firm or organization?	0.156	0.203	0.003	0.032	**−0.783**
How much would you prefer to work in a large firm or organization?	0.192	0.187	0.081	0.130	**0.752**

Rotation method: Varimax. Note: Significant values are in bold. Factors “part time vs. full time” and “organization size” were excluded from the analysis since both had only two questions loaded in them which is not enough for proper reliability.

**Table 3 ijerph-19-09021-t003:** Variable intercorrelations.

Measures	1	2	3	4	5	6	7	8	9	10	11	12
1. Age	-											
2. Gender	−0.034	-										
3. Education	0.400 ***	−0.087	-									
4. SES	−0.167 **	0.039	−0.166 **	-								
5. Work experience	0.360 ***	−0.098	0.242 ***	0.030	-							
6. Self-assessed health	−0.080	0.018	−0.188 ***	0.278 ***	0.036	-						
7. SE	−0.024	0.033	−0.021	0.295 ***	0.143 **	0.347 ***	-					
8. Subjective norms	0.076	−0.051	0.156 ***	0.090	0.125 *	0.075	0.233 ***	-				
9. Attitudes towards Work	−0.011	−0.099	−0.094	0.112 *	0.055	0.085	0.319 ***	0.221 ***	-			
10. Intention to work	0.191 ***	−0.033	0.252 ***	0.023	0.168 **	0.005	0.201 ***	0.362 ***	0.182 ***	-		
11. Interest, security, and advancement at work.	0.077	0.003	0.216 ***	0.128 *	0.081	0.033	0.528 ***	0.350 ***	0.264 ***	0.407 ***	-	
12. Willingness to work in the free market	0.027	−0.054	0.184 ***	0.071	0.112 *	0.068	0.276 ***	0.187 ***	0.084	0.185 ***	0.313 ***	-
Mean	240.83	-	20.83	30.32	20.37	700.78	30.88	40.11	40.01	40.11	40.44	30.55
Standard Deviation	30.44	-	10.05	00.93	10.39	190.83	00.86	00.86	00.60	00.83	00.68	10.19
VIF	10.804	10.835	10.658	10.428	10.923	10.400	10.810	10.497	10.282	-	-	-

Note: Means and SDs for dichotomous variables were omitted, their frequencies are presented in the text. * *p* < 0.05, ** *p* < 0.01, *** *p* < 0.001.

**Table 4 ijerph-19-09021-t004:** The mediation effects of attitude toward work, self-efficacy, and subjective norms on the three outcome variables.

Measures	Direct Effect	Total Effect	Mediation Effect by Sobel’s Z				95% CI	
			Attitude toward Work	Self-Efficacy	Subjective Norms	Self-Assessed Health	Lower Bound	Upper Bound
Dependent variable: Interest, security, and advancement at work								
Independent variables:								
(1) SES	*b* = 0.01, *SE* = 0.03, *ns*	*b* = 0.07, *SE* = 0.03, *p* < 0.05	*-*	4.92 ***	*-*	*-*	0.03	0.12
(2) Education	*b* = 0.09, *SE* = 0.02, *p* < 0.001	*b* = 0.11, *SE* = 0.03, *p* < 0.001	*-*	*-*	2.40 *	*-*	0.002	0.017
Dependent variable: Intention to work								
Independent variable:								
(3) Education	*b* = 0.13, *SE* = 0.03, *p* < 0.001	*b* = 0.18, *SE* = 0.03, *p* < 0.001	*-*	*-*	2.99 **	*-*	0.02	0.08

Note: *p* < 0.06, * *p* < 0.05, ** *p* < 0.01, *** *p* < 0.001.

**Table 5 ijerph-19-09021-t005:** The mediation effect of self-assessed health on self-efficacy.

Measures	Direct Effect	Total Effect	Mediation Effect by *Z* of Sobel	Confidence Interval 95%	
			Self-Assessed Health	Lower Bound	Upper Bound
Dependent variable: self-efficacy					
Independent variables:					
(1) SES	*b* = 0.17, *SE* = 0.04, *p <* 0.001	*b* = 0.23, *SE* = 0.04, *p* < 0.001	Z = 3.90 ***	0.03	0.12

Note: *** *p* < 0.00.

## Data Availability

Upon request.

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
