# Peer review of "Education Makes the Difference: Work Preferences of Young Adults with Physical Disability"

_ijerph, 2022, doi:10.3390/ijerph19159021_

Round 1
Reviewer 1 Report
I think the results are very interesting. There are too many variables and it can be hard to follow. I think it shoud be improved, if for example the introduction is divided in different sections. I suggest the same for the results, there are two many variables, and they could be organized in a different way.
There are some mistakes through the text: to work" Responses were recorded (p.6).
In general, it is a very interesting article.
Table 3 is hard to read. The model has two many variables, may
Author Response
Reviewer 1:
There are too many variables, and it can be hard to follow. I think it should be improved, if for example the introduction is divided in different sections. I suggest the same for the results, there are too many variables, and they could be organized in a different way.
Our response: Thank you for this comment. We have now divided the introduction into three sections and added the hypothesis at the end of each. We have also addressed each hypothesis in the Results.
There are some mistakes through the text: to work" Responses were recorded (p.6)
Our response: We have corrected this mistake and others, thank you.
Table 3 is hard to read. The model has too many variables
Our response: As the Reviewer has noted, the TPB model is complex. This is the first time that a health model has been used to test behavioral intent for work. We have carefully followed some of the variables required by the model itself and for the purposes of this study it is important to emphasize the relevance of the mediation variables of the model itself as well as the mediation variables related to health, such as self-assed health, and the mediation variables of employment, such as work experience.

Reviewer 2 Report
The paper undoubtedly presents very significant and interesting results. I suggest that it would be interesting to look more closely at the influence of socio-economic status, as in this study 55% of the sample is in the upper SES range, which may also determine the results.
Author Response
Reviewer 2:
I suggest that it would be interesting to look more closely at the influence of socio-economic status, as in this study 55% of the sample is in the upper SES range, which may also determine the results.
Our response: Thank you very much for this observation. We have added a sentence regarding this limitation in the Discussion, on p. 11, r. 395.
We have added a sentence referring to this under Limitations and Future Directions, p. 13, r. 487.

Reviewer 3 Report
This study offers important and compelling information regarding preparation for, enthusiasm about, and experience with work among young adults with disability. The authors note the study's shortcomings and encourage others (perhaps even themselves) to investigate readiness for and hopes of employment among younger populations of YAPD. I appreciate the focus on self-efficacy as a marker for likely success in the workforce.
Author Response
Reviewer 3:
This study offers important and compelling information regarding preparation for, enthusiasm about, and experience with work among young adults with disability. The authors note the study's shortcomings and encourage others (perhaps even themselves) to investigate readiness for and hopes of employment among younger populations of YAPD. I appreciate the focus on self-efficacy as a marker for likely success in the workforce.
Our response: Thank you very much for your support.

Reviewer 4 Report
The main drawbacks of the paper are the following:
The Literature review or Background section is missing.
We suggest to authors to separate the Introduction section into two sections: one Introduction and the other devoted to justify the relevance of hypotheses that are not clearly presented, even on p. 3, r. 122-126 is mentioned “We hypothesized that work experience, education and SES would be positively and directly associated with intentions to work and work preferences. We further hypothesized that attitudes toward work, subjective norms and self-efficacy would mediate the link between work experience, SES and 125 education and self-efficacy”.
The hypotheses were not clearly defended/refuted.
The results were not well-presented to readers to understand the focus of the research study.
The results must be interpretive rather than just descriptive and connect the research results with relevant literature citations for validity and reliability.
We suggest correlating the results with the effects on improving existing knowledge.
The discussions are not well-presented and them do not integrate with the results of the research study to provide a coherent scholarly argument.
Conclusion section is missing even some elements related to this section are presented in section Discussion,
Good luck!
Author Response
Reviewer 4:
The Literature review or Background section is missing.
We suggest to authors to separate the Introduction section into two sections: one Introduction and the other devoted to justify the relevance of hypotheses that are not clearly presented, even on p. 3, r. 122-126 is mentioned “We hypothesized that work experience, education and SES would be positively and directly associated with intentions to work and work preferences. We further hypothesized that attitudes toward work, subjective norms and self-efficacy would mediate the link between work experience, SES and education and self-efficacy”.
Our response: Thank you for this comment. See our first answer to Reviewer 1.
The hypotheses were not clearly defended/refuted.
Our response: We have corrected the mistakes in the hypotheses on p. 4, r. 153-159. In addition, we have clarified all the hypotheses.
The results were not well-presented to readers to understand the focus of the research study.The results must be interpretive rather than just descriptive and connect the research results with relevant literature citations for validity and reliability.
Our response: As is customary in quantitative studies and following articles using health models published in the International Journal of Environmental Research and Public Health (IJERPH), there is a separation between the reported findings and the part of the discussion that relates to their interpretation (see, e.g., Zhou et al., 2022). To make things clearer, we have included the hypotheses in the findings and written explicitly whether each has been confirmed or not.
We suggest correlating the results with the effects on improving existing knowledge.
Our response: The correlation between improving existing knowledge and the results is now presented in the Discussion.
The discussions are not well-presented and them do not integrate with the results of the research study to provide a coherent scholarly argument.
Our response: In the Discussion, we have added the hypotheses and discussed each hypothesis. We hope the Discussion is clearer now.
Conclusion section is missing even some elements related to this section are presented in section Discussion
Our response: Thank you for this comment. We have added a Conclusion on p. 13, r. 503-511.

Round 2
Reviewer 4 Report
The paper was improved, but it needs improvements as following:
The Introduction section is underdeveloped and contains elements related to Literature review.
We mention that the reviewer has access to her/his evaluation/answer to it and to to the answer to other evaluators.
The hypotheses are still not clearly defended/refuted.
The discussions are underdeveloped.
The Conclusion section should include the key focus of the study.
The conclusion is not supported by the research data, which does not indicate a clearer path for future studies on the topic.
A follow-up of restate findings with supporting literature reviews could make the conclusion section more effective.
Good luck!
Author Response
Reviewer 4
- The paper was improved, but it needs improvements as following:
- The Introduction section is underdeveloped and contains elements related to Literature review.
Our response: Thank you for your encouragement to correct our study. In the current version we divided the introduction to a short introductory section which included the aims of this paper. Next, we presented the literature review which was also divided as suggested in the previous review. In addition, we added new literature, p. 2 r. 91.
- The hypotheses are still not clearly defended/refuted.
Our response: as suggested in the previous round, as requested by the reviewers, we added aims, and for each aim, the relevant hypothesis. We understand that the reviewers approved this correction. It is important to note that our aims and hypothesis were derived from the TPB model.
- The discussions are underdeveloped.
Our response: Our discussions follow the TPB model. We discussed the results according to empirical previous studies and theoretical conceptualization.
- The Conclusion section should include the key focus of the study.
Our response: In the conclusion section we now added following the reviewer request p. 14 r. 512-530. We expended the conclusion section and added the key focus of the study.
- The conclusion is not supported by the research data, which does not indicate a clearer path for future studies on the topic.
A follow-up of restate findings with supporting literature reviews could make the conclusion section more effective.
Our response: See p. 14 r. 512-530.